

# Assessment of genetic diversity and phylogenetic relationship of local coffee populations in southwestern Saudi Arabia using DNA barcoding

Habib Khemira[1], Mosbah Mahdhi[1,2], Muhammad Afzal[3],
Mohammed D.Y. Oteef[4], Taieb Tounekti[5], Zarraq AL-Faifi[6] and Wail Alsolami[6]

[1] Centre for Environmental Research and Studies, Jazan University, Jazan, Saudi Arabia
[2] Laboratory of Biodiversity and Valorization of Bioresources in Arid Zones, Faculty of Sciences of Gabes, University of Gabes, Gabes, Tunisia
[3] Department of Plant Production, College of Food and Agricultural Sciences, King Saud University, Riyadh, Saudi Arabia
[4] Department of Chemistry, College of Science, Jazan University, Jazan, Saudi Arabia
[5] Laboratory of Process Engineering & Industrial Systems (LR11ES54), National Engineering School of Gabes, University of Gabes, Gabes, Tunisia
[6] Department of Biology, College of Science, Jazan University, Jazan, Saudi Arabia

Corresponding author
Habib Khemira,
habibkhemira@yahoo.com

## ABSTRACT

The genetic diversity of local coffee populations is crucial to breed new varieties better adapted to the increasingly stressful environment due to climate change and evolving consumer preferences. Unfortunately, local coffee germplasm conservation and genetic assessment have not received much attention. Molecular tools offer substantial benefits in identifying and selecting new cultivars or clones suitable for sustainable commercial utilization. New annotation methods, such as chloroplast barcoding, are necessary to produce accurate and high-quality phylogenetic analyses. This study used DNA barcoding techniques to examine the genetic relationships among fifty-six accessions collected from the southwestern part of Saudi Arabia. PCR amplification and sequence characterization were used to investigate the effectiveness of four barcoding loci: atpB-rbcl, trnL-trnF, trnT-trnL, and trnL. The maximum nucleotide sites, nucleotide diversity, and an average number of nucleotide differences were recorded for atpB-rbcl, while trnT-trnL had the highest variable polymorphic sites, segregating sites, and haploid diversity. Among the four barcode loci, trnT-trnL recorded the highest singleton variable sites, while trnL recorded the highest parsimony information sites. Furthermore, the phylogenetic analysis clustered the *Coffea arabica* genotypes into four different groups, with three genotypes (KSA31, KSA38, and KSA46) found to be the most divergent genotypes standing alone in the cluster and remained apart during the analysis. The study demonstrates the presence of considerable diversity among coffee populations in Saudi Arabia. Furthermore, it also shows that DNA barcoding is an effective technique for identifying local coffee genotypes, with potential applications in coffee conservation and breeding efforts.

## INTRODUCTION

Coffee is one of the most commercially significant crops, and the second most traded commodity after oil (*Mussatto et al., 2011*). In addition to its high export value, coffee has also gained in cultural significance over the past few decades. Despite there being more than 125 reported species in the genus *Coffea*, only two species, *Coffea arabica* L. (also known as Arabica coffee) and *C. canephora* Pierre ex A. Froehner (known as Robusta coffee) are grown commercially (*Mishra, 2019*). The total annual global coffee production in 2022 was 10.2 million tons, about 60% of which were Arabica coffee (*USDA, 2023*). Coffee's genetic development is progressing at a sluggish pace despite its enormous economic relevance (*Mishra, 2019*). The collection, characterization, and wise use of accessible germplasm material for any crop plant species contribute to its genetic development and long-term viability (*Nguyen & Norton, 2020*). Therefore, enhancing diversity from both local and foreign sources is critical for the improvement of crops (*Migicovsky et al., 2019*). For historical reasons, the main issue with Arabica coffee has been its narrow genetic base that limits its adaptation to changing environments (*Mishra, 2019*). To get around this problem, breeders made use of wild coffee diploid species to introduce new genes into Arabica genotypes (*Mishra, 2019*). For instance, the leaf rust-resistant Arabica cultivar Timor Hybrid got its resistance from its *C. canephora* parent; it was later used as a parent to develop several new rust-resistant cultivars such as Catimor and Ruiru 11 (*World Coffee Research, 2023*). For bean and liquor quality traits, the wild tetraploid Arabica genotypes from the species' center of origin and the little-known ancient varieties from the Arabian Peninsula offer a wide gene pool to explore (*Montagnon et al., 2021*). Despite the potential importance of coffee heirlooms from the Arabian Peninsula as a source of genetic diversity, there is limited information available on these genotypes. This information is essential for the development of new coffee varieties that can better adapt to changing environmental conditions, increasing pest and disease pressure and changing consumer preferences (*Herrera & Lambot, 2017*). Furthermore, since over 60% of wild coffee species are in danger of extinction due to accelerated environmental change, gathering complete information and characterizing this germplasm is of utmost importance (*Davis et al., 2019*).

Another issue facing the coffee industry as it struggles to cope with an over-supplied market is adulteration. It has long been known that coffee is often adulterated with less expensive and readily available plant material (*Oliveira & Franca, 2015*). Coffee adulteration has become a more serious issue for the industry in recent years due to the significant expansion in the variety of coffee recipes, stores, and ultimately consumers (*Choudhary et al., 2020*). Therefore, developing molecular means like genetic barcodes to identify and authenticate the varieties can help mitigate the problem.

In Saudi Arabia and Yemen, *C. arabica* has been cultivated for at least four centuries on the terraced slopes and narrow valleys of the western mountains at different altitudes ranging mostly from 1200 to 2000 m above sea level (a.s.l.) (*Al-Zaidi et al., 2016*; *Al-Asmari, Zeid & Al-Attar, 2020*). Most of what is grown now in southwestern Saudi Arabia are old cultivars that have been around for hundreds of years (*Tounekti et al., 2017*). It is likely

that these diverse populations are a result of successive introductions of genetic material from Eastern Ethiopia by Arab traders over centuries of uninterrupted exchange across the narrow strait of Bab El-Mandeb (*Montagnon et al., 2022*). Therefore, it is safe to assume that the southwestern corner of the Arabian Peninsula contains the most genetic diversity of *C. arabica* outside the species' center of origin in the Ethiopian highlands (*Montagnon et al., 2021*). Regrettably, the scientific community has shown only limited interest in these genetic resources, with the notable exception being the 1989 FAO expedition to southern Yemen (*Eskes, 1989*) and three subsequent studies (*Tounekti et al., 2017*; *Montagnon et al., 2021*; *Al-Ghamedi et al., 2023*). These studies reported the existence of considerable diversity among coffee populations in the Arabian Peninsula. It is worth noticing that the present coffee populations have evolved over hundreds of years in a semi-arid environment (*De Pauw, 2002*) marked by recurring droughts, uneven distribution of rainfall, heat stress and high irradiance. Therefore, it is expected that these genotypes could be the source of interesting genes that confer stress tolerance (*Tounekti et al., 2018*).

In recent years, DNA metabarcoding has emerged as a progressive alternative approach enabling qualitative analysis (species or genus identification for certain taxa) and to some extent, quantitative analysis of complex biological mixtures. This method utilizes high-throughput sequencing (HTS) and comparative analysis of specific DNA sequences known as "DNA barcodes" to differentiate the species present within the mixture (*Omelchenko et al., 2022*). One of the main challenges in plant barcoding is the selection of an appropriate DNA barcode for the target taxa (*Coissac, Riaz & Puillandre, 2012*; *Taylor & Harris, 2012*). The effectiveness of the primary chloroplast markers, initially suggested by the CBOL group to consist of matK and rbcL, is a crucial factor to consider in this context. The same study also demonstrated that the trnL marker reliably identifies 50% of the plant species considered, affirming its credibility as a taxonomic tool for plant identification (*Valentini et al., 2009*).

The difference among the coffee species have been established based on phylogenetic analysis using different barcode intergenic spacer sequences (*Cros et al., 1998*; *Tesfaye et al., 2014*), introns (*Tesfaye et al., 2007*), plastid DNA, and internal transcribed spacer (ITS) region of rDNA (*Lashermes et al., 1997*), and different combination of four plastid and ITS primers (*Jingade et al., 2019*). Similarly, the chloroplast DNA (cpDNA) sequence variation is also widely used for identification and for making phylogenetic inferences at different taxonomic levels (*Li et al., 2019*). Introns and intergenic spacers are known to exhibit high rates of mutation (*Barakat et al., 2010*). The trnT-trnL, trnL-trnF and atpB-rbcL intergenic spacers, the trnL intron region were successfully used for species identification at low taxonomic levels. These regions also have been used in phylogenetic studies to figure out the cytoplasmic differences as well as the demographic history of several species (*Barakat et al., 2010*; *Mashaly et al., 2017*). These markers were successfully used for the identification of species and the construction of phylogenies at different taxonomic levels within the *Rubiaceae* family (*Kårehed et al., 2008*; *Ginter, Razafimandimbison & Bremer, 2015*). Therefore, these four barcode loci were used for the identification of local coffee genotypes present in the southwestern region of Saudi Arabia.

Overall, further research is necessary to fully comprehend the diversity and potential of diploid and tetraploid coffee species and to utilize this information to develop new coffee varieties that can better meet the needs of farmers and consumers in the future. The present study aims to use DNA barcoding to identify the local coffee genotypes in Saudi Arabia, to estimate the genetic diversity of the local coffee populations and to examine their genetic relatedness using chloroplast intergenic spacer markers.

## MATERIALS AND METHODS

### Plant material

The plant material for the study was collected as previously described by *Al-Ghamedi et al. (2023)*. A survey was carried out at several sites in the Sarawat mountain range, running parallel to the Red Sea from the southeast to the northwest through the three administrative regions of Jazan, Assir, and Al-Baha. The survey covered a narrow strip of terraced mountains located between latitudes 17 °N and 20 °N, the most northern location where coffee is commercially grown in the world. The coffee gardens included in the survey were found at altitudes ranging from 900 to 2000 m a.s.l. In total, we collected young leaves from 56 accessions, from Jebel Fayfa (Fayfa district), Eddayer, Maadi (Haroub district), Jebel Al-Gahr (Al-Rayth district), Rayda valley (Assouda district in Assir region), Mahayel Assir district, Al-Majarda district and Jebel Shada (Al-Mekhwah district of Al-Baha region) (Table 1). We tagged and sampled 3-4 trees representing each tree population. Each accession was given a code starting with the acronym "KSA" (*e.g.*, KSA-1), but, for the sake of simplicity, we dropped the acronym in the figures. The letter "R" was added to the code of accessions 1–19, 45, and 51 to indicate that they were sourced from a small, local coffee germplasm collection established in the Fayfa district.

### DNA extraction

Portions of this text were previously published as part of a preprint (*Khemira et al., 2023*). Plant material, consisting of young leaves from various *C. arabica* accessions, was collected from representative trees in each population, transported to the laboratory in a storage container and stored at −20 °C prior to DNA extraction. The leaves were sanitized by immersing them in a 5% sodium hypochlorite solution for 1–2 min and then rinsing them with sterile distilled water. The material was then ground in liquid nitrogen and stored in an −80 °C freezer. DNA was extracted from 100 mg of mixed powder using an innuPREP Plant DNA Kit (Analytik, Jena, Germany), following the manufacturer's protocol. DNA quality and concentration were determined using a Nanodrop ND-1000 spectrophotometer (Saveen Werner, Limhamn, Sweden).

### Chloroplastic DNA amplification and sequencing

Four chloroplast DNA regions were considered (Table 2). PCR was performed in a 25 µl volume containing 2 µl of template DNA, 10 µl of 1X innuMix Standard PCR, and 1 µM of each primer (Table 2) (*Khemira et al., 2023*). The Gene Amp PCR System 9700 was used with the following program: initial denaturation at 94 °C for 5 min, 35 cycles of denaturation at 94 °C for 1 min, annealing at 49–52 °C for 60–75 s, and elongation at

**Table 1   Altitude and latitude of the sites where the coffee accessions were sourced.** The sites are located between longitudes 42°22′ and 43°07′E.

| # | Accession no. | Region | District | Altitude (m a.s.l.) | Latitude |
|---|---|---|---|---|---|
| 1 | KSA1R | Jazan | Khacher/Al-Zoughli | 1,254 | 17°18′03″N |
| 2 | KSA2R | Jazan | Khacher/Al-Guatil | 1,484 | 17°19′01″N |
| 3 | KSA3R | Jazan | Khacher/Al-Guatil | 1,484 | 17°19′01″N |
| 4 | KSA4R | Jazan | Jebel Fayfa | 1,541 | 17°15′21″N |
| 5 | KSA5R | Jazan | Wadi Dafa | 1,254 | 17°25′41″N |
| 6 | KSA6R | Jazan | Tallan | 1,672 | 17°23′12″N |
| 7 | KSA7R | Jazan | Tallan | 1,672 | 17°23′12″N |
| 8 | KSA8R | Jazan | Tallan | 1,546 | 17°23′01″N |
| 9 | KSA9R | Jazan | Tallan | 1,672 | 17°23′12″N |
| 10 | KSA10R | Jazan | Khacher/Al-Zoughli | 1,254 | 17°18′03″N |
| 11 | KSA11R | Assir | Rayda | 1,594 | 18°11′37″N |
| 12 | KSA12R | Jazan | Maaddi | 1,287 | 17°29′29″N |
| 13 | KSA13R | Jazan | Maaddi | 1,344 | 17°29′29″N |
| 14 | KSA15R | Al-Baha | Shada Al-ala | 1,548 | 19°50′54″N |
| 15 | KSA16R | Assir | Rayda | 1,594 | 18°11′37″N |
| 16 | KSA17R | Assir | Rayda | 1,519 | 18°11′37″N |
| 17 | KSA18R | Assir | Al-Majarda | 1,329 | 19°09′35″N |
| 18 | KSA19R | Assir | Al-Majarda | 1,300 | 19°09′35″N |
| 19 | KSA20 | Jazan | Jebel Fayfa | 1,260 | 17°15′20″N |
| 20 | KSA21 | Jazan | Jebel Fayfa | 1,260 | 17°15′20″N |
| 21 | KSA22 | Jazan | Jebel Fayfa | 1,260 | 17°15′20″N |
| 22 | KSA23 | Jazan | Jebel Fayfa | 1,260 | 17°15′20″N |
| 23 | KSA24 | Jazan | Jebel Fayfa | 1,260 | 17°15′20″N |
| 24 | KSA25 | Jazan | Jebel Fayfa | 1,260 | 17°15′20″N |
| 25 | KSA26 | Jazan | Jebel Fayfa | 1,550 | 17°15′24″N |
| 26 | KSA27 | Jazan | Jebel Fayfa | 1,550 | 17°15′24″N |
| 27 | KSA28 | Jazan | Jebel Fayfa | 1,550 | 17°15′24″N |
| 28 | KSA29 | Jazan | Al-Gahr | 1,846 | 17°38′08″N |
| 29 | KSA30 | Jazan | Al-Gahr | 1,846 | 17°38′08″N |
| 30 | KSA31 | Jazan | Al-Gahr | 1,846 | 17°38′08″N |
| 31 | KSA32 | Jazan | Al-Gahr | 1,846 | 17°38′08″N |
| 32 | KSA33 | Jazan | Al-Gahr | 1,846 | 17°38′08″N |
| 33 | KSA34 | Jazan | Jebel Fayfa | 1,660 | 17°15′55″N |
| 34 | KSA35 | Jazan | Jebel Fayfa | 1,660 | 17°15′55″N |
| 35 | KSA36 | Jazan | Jebel Fayfa | 1,450 | 17°15′59″N |
| 36 | KSA37 | Jazan | Eddayer | 1,100 | 17°22′10″N |
| 37 | KSA38 | Jazan | Eddayer | 1,228 | 17°22′10″N |
| 38 | KSA39 | Jazan | Eddayer | 1,228 | 17°22′10″N |
| 39 | KSA40 | Jazan | Haroub | 1,100 | 17°29′29″N |
| 40 | KSA41 | Assir | Rayda | 1,450 | 18°11′37″N |

*(continued on next page)*

**Table 1** (*continued*)

| # | Accession no. | Region | District | Altitude (m a.s.l.) | Latitude |
|---|---|---|---|---|---|
| 41 | KSA42 | Assir | Rayda | 1,450 | 18°11′37″N |
| 42 | KSA43 | Assir | Rayda | 1,400 | 18°11′37″N |
| 43 | KSA44 | Jazan | Jebel Fayfa | 1,524 | 17°15′48″N |
| 44 | KSA45R | Jazan | Jebel Fayfa | 1,524 | 17°15′48″N |
| 45 | KSA46 | Jazan | Al-Gahr | 1,750 | 17°39′01″N |
| 46 | KSA47 | Jazan | Al-Gahr | 1,750 | 17°39′01″N |
| 47 | KSA48 | Jazan | Jebel Fayfa | 1,260 | 17°15′20″N |
| 48 | KSA49 | Jazan | Jebel Fayfa | 1,260 | 17°15′20″N |
| 49 | KSA50 | Jazan | Jebel Fayfa | 1,260 | 17°15′20″N |
| 50 | KSA51R | Jazan | Jebel Fayfa | 1,524 | 17°17′13″N |
| 51 | KSA52 | Jazan | Jebel Fayfa | 1,550 | 17°15′24″N |
| 52 | KSA59 | Assir | Al-Majarda | 1,329 | 19°09′35″N |
| 53 | KSA60 | Assir | Al-Majarda | 1,300 | 19°09′35″N |
| 54 | KSA61 | Al-Baha | Shada Al-ala | 1,548 | 19°50′54″N |
| 55 | KSA62 | Al-Baha | Shada Al-ala | 1,548 | 19°50′54″N |
| 56 | KSA63 | Al-Baha | Shada Al-ala | 1,548 | 19°50′54″N |

**Table 2  General information about the PCR primers used in this study.**

| Sr# | Sequence 5′-3′ | Target | PCR condition | Source |
|---|---|---|---|---|
| 1 | CATTACAAATGCGATGCTCT TCTACCGATTTCGCCATATC | trnT-trnL | Hybridation: 50 °C/1 min Elongation: 72 °C/1min | [29] |
| 2 | CGAAATCGGTAGACGCTACG GGGGATAGAGGGACTTGAAC | trnL | Hybridation: 49 °C/1.15 min Elongation: 72 °C/1.15 min | [29] |
| 3 | GGTTCAAGTCCCTCTATCCC ATTTGAACTGGTGACACGAG | TrnL-trnF | Hybridation:52 °C/1 min Elongation: 72 °C/1min | [29] |
| 4 | GAAGTAGTAGGATTGATTCTC TACAGTTGTCCATGTACCAG | atpB-rbcL | Hybridation: 50 °C/1 min Elongation: 72 °C/1min | [30] |

72 °C for 60–75 s, followed by a final polymerization at 72 °C for 10 min. To check the effectiveness of PCR, positive control using sterile water was included in all amplifications. The PCR products were checked by electrophoresis on 1% agarose gel in TAE buffer, and DNA was visualized under UV light after staining with ethidium bromide.

The amplified products were purified using the GFX PCR kit (GE Healthcare, Chicago, IL, USA). Sequencing reactions were carried out by Congenic using Sanger technology, separately for each strand to obtain independent forward and reverse sequences. The forward and reverse fragments were aligned, and additional reactions were conducted in case of any discrepancies.

## Sequence analysis

The scanner software-2 was utilized to determine the quality of the sequences. The four barcode samples of each *C. arabica* genotype were manually curated and aligned using the Contig assembly program in Bio Edit 7.0 software to ensure high-quality sequences.

Nucleotide sequences obtained from the 57 accessions were initially aligned using CLUSTAL W (*Thompson, Higgins & Gibson, 1994*) and analyzed with MEGA program version X. The number of individuals, number of nucleotide sites, variable polymorphic sites, number of segregating sites, number of haplotypes, nucleotide diversity, and average number of nucleotide differences of each barcode marker and consensus sequence were measured using DNAsp (v6) (*Rozas et al., 2019*). The quantification of insertion events in the sequence was determined by the number of variable sites where the addition of one or more nucleotides signals polymorphism. Likewise, the number of deletions was determined by the variable sites where polymorphism arises due to the removal of one or more nucleotides. The identification of the number of transitions in the sequences was based on the number of variable sites where polymorphism occurred due to the exchange between two purines (A and G) or two pyrimidines (C and T). On the other hand, the number of transversions was determined by the variable sites where polymorphism resulted from the replacement of a purine with a pyrimidine. To determine the number of mutation events that have occurred in a sequence, the sum of variable sites and the number of distinct mutations observed at the same nucleotide site across different samples are combined. This quantification considers both different types of polymorphisms and multiple occurrences of mutations within the sequence. Various parameters were estimated for each sequence region to differentiate them, based on the number of monomorphic or polymorphic sites, the number of parsimony informative sites (PIS), nucleotide diversity ($\pi$), haplotype diversity (Hd), and the total number of mutations (*Hosein et al., 2017*; *Rabaan et al., 2020*), singleton variable site (STVC) (*Pettengill & Neel, 2010*). The percentage of polymorphic sites for each sequence was determined by dividing the number of variable nucleotides by the length of the entire region and multiplying the result by 100 (*Chen et al., 2023*).

**Evolutionary analysis by Maximum Likelihood method**

The Maximum Likelihood method and the Kimura 2-parameters model proposed by *Kimura (1980)* were used to assess the evolutionary relationships among the genotypes. The tree with the highest log likelihood (−22360.57) is shown. The Neighbor-Join and BioNJ algorithms were applied to a matrix of pairwise distances obtained using the Maximum Composite Likelihood approach to obtain the initial tree for the heuristic search. The topology with the superior log likelihood value was retained. The tree was drawn to scale with the length of the branches proportional to the number of substitutions per site. This analysis involved 57 nucleotide sequences. There was a total of 5381 diverse positions present in the final dataset. Evolutionary analyses were conducted in MEGA X (*Kumar et al., 2018*).

## RESULTS

The successful amplification of all four intergenic spacer barcode sequences (atpB-rbcl, TrnT-trnL, TrnL-trnF, TrnL) was achieved, resulting in a single band of the expected size. The respective sequences for each barcode were submitted to the National Center for Biotechnology Information (NCBI) *via* Bankit submission. The accession number of each barcode for the 56 *C. arabica* genotypes is presented in Table 3. All genotypes were

identified as *C. arabica* for all barcodes except KSA2R, KSA41, KSA42, and KSA43 for primer atpB-rbcl.

The number of nucleotide sites (NNS), variable polymorphic sites (VPS), number of segregating sites (NSS), number of haplotypes (NH), nucleotide diversity (ND), and average number of nucleotide differences(ANND) for each barcode primer and the cumulative results for all four primers (Table 4). The combined sequences showed the highest NNS (4114), followed by the atpB-rbcl primer, while the trnL primer had the lowest NNS. The trnT-trnL primer had the highest number of variable polymorphic sites VPS (341), followed by atpB-rbcl, while the lowest (154) was recorded for TrnL-trnF. The combined sequences had the highest number of segregating sites (NSS) followed by the trnT-trnL primer, while trnL and trnL-trnF had the lowest number. The number of haplotypes was highest for trnT-trnL and lowest for trnL-trnF and atpB-rbcl while trnL and the combination of all four markers were intermediate. The primer atpB-rbcl had the highest ND, followed by trnT-trnL (0.051), with TrnL-trnF showing the lowest ND. Additionally, the atpB-rbcl had the highest ANND (185.54), while TrnL-trnF exhibited the lowest value (25.23) for ANND.

The nucleotide base composition of each barcode primer was determined and is presented in Table 5. The average nucleotide base composition of atpB-rbcl was recorded as 33.15% T(U), 16.60% C, 34.49% A, and 15.76% G. For trnL, the composition was 26.7% T(U), 15.9% C, 37.6% A, and 19.8% G. trnT-trnL had a composition of 39.18% T(U), 13.88% C, 33.84% A, and 13.10% G. For trnL-trnF the composition was 32.83% T(U), 19.81% C, 32.21% A, and 15.13% G (Table 5). The singleton variable sites (STVS) and parsimony information sites (PIS) for each chloroplast barcode are presented in Table S1. The trnT-trnL barcode recorded the highest number of STVS (338), followed by trnL-trnF (133), trnL (52), then atpB-rbcl which had the lowest number (1). Similarly, for grand total of PIS was 182 for trnL, 155 for trnT-trnL, 137 for atpB-rbcl and 45 for TrnL-trnF (Table S1). A phylogenetic analysis was constructed based on the concatenated sequences of all four barcode primers using the maximum likelihood method and Kimura 2-parameters model (Fig. 1). The percentage of trees in which the associated taxa clustered together is shown next to the branches. This analysis involved 56 nucleotide sequences, and the final dataset comprised 4,114 positions. The tree is drawn to scale, with branch lengths measured in the number of substitutions per site. The final phylogenetic tree divided the 56 accessions into four distinct groups. The first group contained six accessions (KSA42, KSA29, KSA2R, KSA41, KSA43, and KSA11R) that were mostly from the Rayda district of Assir region. The second group contained seven accessions (KSA51R, KSA3R, KSA27, KSA60, KSA4R, KSA7R, and KSA1R), all from the Jazan Region except KSA60 was from Assir. The third group was formed of 12 accessions (KSA45R, KSA13R, KSA39, KSA25, KSA35, KSA59, KSA52, KSA36, KSA24, KSA22, KSA37 and KSA46), all collected from the Jazan Region except KSA59 from the north of Assir Region. The fourth and largest group contained 43 accessions that can be further subdivided into four subgroups. The first subgroup (IVa) was a diverse one and contained 12 accessions originating from the three regions. Subgroup IVb contained three accessions (KSA33, KSA28 and KSA5R) all from the Jazan Region. Subgroup IVc contained seven accessions, six from Jazan and one
**Table 3** Accession numbers of four barcode primers of 56 *Coffea arabica* genotypes.

| Genotype ID | atpB-rbcl | trnL-trnF | trnT-trnL | trnL |
|---|---|---|---|---|
| KSA1R | OQ718327 | OQ914867 | OQ914923 | OQ953999 |
| KSA2R | – | OQ914868 | OQ914924 | OQ954000 |
| KSA3R | OQ844066 | OQ914869 | OQ914925 | OQ954001 |
| KSA4R | OQ914863 | OQ914870 | OQ914926 | OQ954002 |
| KSA5R | OQ914864 | OQ914871 | OQ914927 | OQ954003 |
| KSA6R | OQ914865 | OQ914872 | OQ914928 | OQ954004 |
| KSA7R | OQ914866 | OQ914873 | OQ914929 | OQ954005 |
| KSA8R | OQ850301 | OQ914874 | OQ914930 | OQ954006 |
| KSA9R | OQ850302 | OQ914875 | OQ914931 | OQ954007 |
| KSA10R | OQ850303 | OQ914876 | OQ914932 | OQ954008 |
| KSA11R | OQ850304 | OQ914877 | OQ914933 | OQ954009 |
| KSA12R | OQ850305 | OQ914878 | OQ914934 | OQ954010 |
| KSA13R | OQ850306 | OQ914879 | OQ914935 | OQ954011 |
| KSA15R | OQ851715 | OQ914880 | OQ914936 | OQ954012 |
| KSA16R | OQ851716 | OQ914881 | OQ914937 | OQ954013 |
| KSA17R | OQ851717 | OQ914882 | OQ914938 | OQ954014 |
| KSA18R | OQ851718 | OQ914883 | OQ914939 | OQ954015 |
| KSA19R | OQ851719 | OQ914884 | OQ914940 | OQ954016 |
| KSA20 | OQ851720 | OQ914885 | OQ914941 | OQ954017 |
| KSA21 | OQ872544 | OQ914886 | OQ914942 | OQ954018 |
| KSA22 | OQ872545 | OQ914887 | OQ914943 | OQ954019 |
| KSA23 | OQ872546 | OQ914888 | OQ914944 | OQ954020 |
| KSA24 | OQ872547 | OQ914889 | OQ914945 | OQ954021 |
| KSA25 | OQ872548 | OQ914890 | OQ914946 | OQ954022 |
| KSA26 | OQ872549 | OQ914891 | OQ914947 | OQ954023 |
| KSA27 | OQ872550 | OQ914892 | OQ914948 | OQ954024 |
| KSA28 | OQ872551 | OQ914893 | OQ914949 | OQ954025 |
| KSA29 | OQ872552 | OQ914894 | OQ914950 | OQ954026 |
| KSA30 | OQ872553 | OQ914895 | OQ914951 | OQ954027 |
| KSA31 | OQ872554 | OQ914896 | OQ914952 | OQ954028 |
| KSA32 | OQ872555 | OQ914897 | OQ914953 | OQ954029 |
| KSA33 | OQ872556 | OQ914898 | OQ914954 | OQ954030 |
| KSA34 | OQ872557 | OQ914899 | OQ914955 | OQ954031 |
| KSA35 | OQ872558 | OQ914900 | OQ914956 | OQ954032 |
| KSA36 | OQ872559 | OQ914901 | OQ914957 | OQ954033 |
| KSA37 | OQ872560 | OQ914902 | OQ914958 | OQ954034 |
| KSA38 | OQ872561 | OQ914903 | OQ914959 | OQ954035 |
| KSA39 | OQ872562 | OQ914904 | OQ914960 | OQ954036 |
| KSA40 | OQ872563 | OQ914905 | OQ914961 | OQ954037 |
| KSA41 | – | OQ914906 | OQ914962 | OQ954038 |
| KSA42 | – | OQ914907 | OQ914963 | OQ954039 |

**Table 3** (*continued*)

| Genotype ID | atpB-rbcl | trnL-trnF | trnT-trnL | trnL |
|---|---|---|---|---|
| KSA43 | – | OQ914908 | OQ914964 | OQ954040 |
| KSA44 | OQ852764 | OQ914909 | OQ914965 | OQ954041 |
| KSA45R | OQ852765 | OQ914910 | OQ914966 | OQ954042 |
| KSA46 | OQ852766 | OQ914911 | OQ914967 | OQ954043 |
| KSA47 | OQ852767 | OQ914912 | OQ914968 | OQ954044 |
| KSA48 | OQ852768 | OQ914913 | OQ914969 | OQ954045 |
| KSA49 | OQ852769 | OQ914914 | OQ914970 | OQ954046 |
| KSA50 | OQ852770 | OQ914915 | OQ914971 | OQ954047 |
| KSA51R | OQ852771 | OQ914916 | OQ914972 | OQ954048 |
| KSA52 | OQ852772 | OQ914917 | OQ914973 | OQ954049 |
| KSA59 | OQ852773 | OQ914918 | OQ914974 | OQ954050 |
| KSA60 | OQ852774 | OQ914919 | OQ914975 | OQ954051 |
| KSA61 | OQ852775 | OQ914920 | OQ914976 | OQ954052 |
| KSA62 | OQ852776 | OQ914921 | OQ914977 | OQ954053 |
| KSA63 | OQ852777 | OQ914922 | OQ914978 | OQ954054 |

**Notes.**

KSA2; KSA41; KSA42; KSA43 were not identified in the database for atpB-rbcl barcode.

**Table 4** Summary of nucleotide sites, variable polymorphic sites, number of segregating sites, haploid diversity, nucleotide diversity, and average number of nucleotide difference.

| Barcode name | Individual | NNS | VPS | NSS | NH | ND | ANND |
|---|---|---|---|---|---|---|---|
| **atpB-rbcl** | 56 | 1,139 | 341 | 341 | 17 | 0.54 | 185.54 |
| **trnL** | 55 | 551 | 237 | 237 | 31 | 0.056 | 18.93 |
| **trnL-trnF** | 56 | 1,055 | 154 | 154 | 17 | 0.046 | 25.23 |
| **trnT-trnL** | 56 | 988 | 421 | 421 | 50 | 0.051 | 40.50 |
| **atpB-rbcl+trnL+trnL-trnF+trnT-trnL** | 223 | 4,114 | 651 | 651 | 37 | 0.11 | 295 |

**Notes.**

NNS, Number of nucleotide sites; VPS, variable polymorphic sites; NSS, number of segregating sites; NH, number of haplotypes; ND, Nucleotide diversity; ANND, average number of nucleotide difference.

from Al-Baha. Subgroup IVd contained 11 accessions, eight from the Jazan region, two from Assir and one from Al-Baha.

## DISCUSSION

The genetic diversity present in any crop wild or primitive relatives plays a crucial role in the effectiveness of crop improvement programs. These wild or unknown genotypes exist in diverse habitats, many of which are currently facing significant threats due to habitat degradation and climate change (*Davis et al., 2019*). Therefore, developing molecular means like the genetic barcodes used to identify and validate the coffee varieties can help mitigate the problem.

In Saudi Arabia and Yemen, *C. arabica* has been cultivated for at least four centuries on the terraced slopes and narrow valleys of the western mountains at altitudes ranging mostly from 1,200 to 2,000 m above sea level (a.s.l.) (*Al-Zaidi et al., 2016*; *Al-Asmari, Zeid & Al-Attar, 2020*).

**Table 5  Nucleotide base substitution matrix of four barcoding markers in *Arabica coffee*.**

| Genotypes | atpb–rbcl | | | | | TrnL–TrnF | | | | | TrnL | | | | | TrnT–TrnL | | | | |
|---|---|---|---|---|---|---|---|---|---|---|---|---|---|---|---|---|---|---|---|---|
| | T(U) | C | A | G | Total | T(U) | C | A | G | Total | T(U) | C | A | G | Total | T(U) | C | A | G | Total |
| KSA1R | 36.36 | 16.01 | 30.48 | 17.15 | 968 | 32.94 | 20.38 | 32.46 | 14.22 | 422 | 26.9 | 15.7 | 37.9 | 19.6 | 562 | 39.48 | 13.49 | 33.79 | 13.24 | 808 |
| KSA2R | | | | | | 33.25 | 19.21 | 33.50 | 14.04 | 406 | 28.5 | 15.5 | 35.8 | 20.2 | 1017 | 39.58 | 14.02 | 33.75 | 12.66 | 806 |
| KSA3R | 35.60 | 15.89 | 31.48 | 17.03 | 969 | 33.25 | 19.76 | 32.53 | 14.46 | 415 | 27.3 | 15.5 | 38.0 | 19.3 | 561 | 39.14 | 13.83 | 34.20 | 12.84 | 810 |
| KSA4R | 35.62 | 15.86 | 31.12 | 17.40 | 977 | 33.74 | 19.56 | 32.03 | 14.67 | 409 | 26.9 | 15.6 | 38.0 | 19.5 | 558 | 36.78 | 14.23 | 33.72 | 15.28 | 949 |
| KSA5R | 35.34 | 15.89 | 31.57 | 17.21 | 982 | 33.58 | 20.34 | 31.86 | 14.22 | 408 | 27.6 | 15.7 | 37.7 | 19.0 | 562 | 37.93 | 14.81 | 33.60 | 13.67 | 878 |
| KSA6R | 34.38 | 15.57 | 31.90 | 18.15 | 1047 | 33.66 | 19.85 | 32.20 | 14.29 | 413 | 27.9 | 16.1 | 36.9 | 19.1 | 559 | 39.63 | 13.46 | 34.69 | 12.22 | 810 |
| KSA7R | 34.34 | 16.49 | 31.90 | 17.27 | 1025 | 32.94 | 19.81 | 32.94 | 14.32 | 419 | 26.4 | 16.3 | 38.0 | 19.3 | 569 | 39.15 | 13.66 | 34.88 | 12.32 | 820 |
| KSA8R | 35.20 | 16.02 | 30.92 | 17.86 | 980 | 33.33 | 20.29 | 31.88 | 14.49 | 414 | 26.4 | 15.5 | 38.4 | 19.7 | 549 | 38.00 | 13.65 | 34.47 | 13.88 | 850 |
| KSA9R | 35.79 | 15.79 | 30.97 | 17.44 | 975 | 27.97 | 17.72 | 34.27 | 20.05 | 429 | 26.4 | 15.6 | 38.3 | 19.6 | 556 | 38.63 | 13.92 | 33.18 | 14.27 | 862 |
| KSA10R | 35.35 | 16.14 | 31.14 | 17.37 | 973 | 30.37 | 22.73 | 32.64 | 14.26 | 484 | 26.3 | 15.6 | 38.1 | 20.0 | 551 | 38.31 | 14.24 | 32.54 | 14.92 | 885 |
| KSA11R | 31.89 | 17.34 | 35.36 | 15.41 | 1038 | 32.39 | 22.98 | 29.76 | 14.88 | 457 | 25.9 | 16.5 | 37.3 | 20.3 | 557 | 36.42 | 13.22 | 31.97 | 18.39 | 832 |
| KSA12R | 32.69 | 16.04 | 36.33 | 14.93 | 991 | 34.00 | 18.60 | 32.00 | 15.40 | 500 | 27.0 | 15.7 | 38.0 | 19.3 | 548 | 39.53 | 13.84 | 33.54 | 13.09 | 802 |
| KSA13R | 33.13 | 16.36 | 35.60 | 14.92 | 972 | 33.25 | 20.15 | 31.80 | 14.81 | 412 | 26.7 | 16.1 | 37.5 | 19.7 | 554 | 40.08 | 13.94 | 34.17 | 11.81 | 796 |
| KSA15R | 32.80 | 16.10 | 36.22 | 14.89 | 994 | 33.66 | 20.34 | 31.72 | 14.29 | 413 | 27.0 | 15.4 | 37.9 | 19.7 | 544 | 39.85 | 14.25 | 33.92 | 11.98 | 793 |
| KSA16R | 33.13 | 16.26 | 35.38 | 15.24 | 978 | 32.89 | 20.00 | 32.00 | 15.11 | 450 | 26.7 | 16.4 | 37.5 | 19.4 | 566 | 39.43 | 14.18 | 33.58 | 12.81 | 804 |
| KSA17R | 32.66 | 16.13 | 36.19 | 15.02 | 992 | 33.98 | 19.28 | 32.05 | 14.70 | 415 | 26.8 | 15.5 | 37.6 | 20.1 | 548 | 39.87 | 13.96 | 33.71 | 12.45 | 795 |
| KSA18R | 32.49 | 16.05 | 35.81 | 15.66 | 1022 | 33.09 | 20.19 | 31.63 | 15.09 | 411 | 26.9 | 15.8 | 37.7 | 19.6 | 551 | 40.30 | 14.23 | 33.88 | 11.59 | 794 |
| KSA19R | 32.79 | 15.94 | 36.14 | 15.13 | 985 | 31.80 | 19.80 | 32.00 | 16.40 | 500 | 26.8 | 15.6 | 37.9 | 19.7 | 557 | 39.53 | 14.00 | 34.20 | 12.27 | 807 |
| KSA20 | 32.40 | 15.90 | 36.40 | 15.30 | 1000 | 33.41 | 19.95 | 31.73 | 14.90 | 416 | 26.1 | 15.4 | 38.3 | 20.1 | 566 | 38.99 | 13.74 | 33.79 | 13.49 | 808 |
| KSA21 | 33.16 | 15.76 | 36.23 | 14.84 | 977 | 30.93 | 19.77 | 34.19 | 15.12 | 430 | 26.1 | 15.3 | 38.2 | 20.4 | 555 | 39.88 | 14.00 | 33.50 | 12.63 | 800 |
| KSA22 | 32.99 | 16.55 | 35.65 | 14.81 | 979 | 32.49 | 19.22 | 33.18 | 15.10 | 437 | 26.6 | 15.0 | 38.4 | 20.0 | 515 | 39.88 | 14.00 | 33.75 | 12.38 | 800 |
| KSA23 | 33.07 | 15.95 | 35.80 | 15.18 | 1028 | 32.85 | 20.05 | 31.64 | 15.46 | 414 | 27.1 | 15.4 | 37.9 | 19.6 | 565 | 39.88 | 14.00 | 33.38 | 12.75 | 800 |
| KSA24 | 32.40 | 16.15 | 35.95 | 15.50 | 929 | 33.50 | 20.39 | 31.55 | 14.56 | 412 | 26.4 | 14.9 | 38.2 | 20.5 | 523 | 39.48 | 13.82 | 33.62 | 13.08 | 803 |
| KSA25 | 32.97 | 15.55 | 36.22 | 15.26 | 1016 | 28.67 | 14.00 | 34.89 | 22.44 | 450 | 26.5 | 16.6 | 37.0 | 19.9 | 548 | 39.88 | 14.00 | 33.63 | 12.50 | 800 |
| KSA26 | 32.90 | 15.33 | 36.45 | 15.33 | 1070 | 34.12 | 19.43 | 31.75 | 14.69 | 422 | 26.6 | 15.8 | 37.9 | 19.7 | 549 | 40.03 | 13.80 | 33.75 | 12.42 | 797 |
| KSA27 | 31.87 | 15.99 | 36.05 | 16.09 | 957 | 33.65 | 19.47 | 32.21 | 14.66 | 416 | 27.2 | 15.7 | 37.4 | 19.7 | 548 | 39.50 | 14.13 | 33.75 | 12.63 | 800 |
| KSA28 | 32.47 | 16.80 | 35.57 | 15.15 | 970 | 32.44 | 19.11 | 32.00 | 16.44 | 450 | 26.2 | 16.3 | 36.4 | 21.1 | 583 | 39.63 | 14.00 | 33.75 | 12.63 | 800 |
| KSA29 | 35.42 | 25.33 | 21.87 | 17.38 | 1070 | 33.57 | 19.08 | 31.40 | 15.94 | 414 | 25.7 | 16.8 | 36.0 | 21.5 | 600 | 39.15 | 14.09 | 33.29 | 13.47 | 802 |
| KSA30 | 32.48 | 16.45 | 36.16 | 14.91 | 979 | 32.77 | 19.76 | 32.77 | 14.70 | 415 | 26.0 | 15.6 | 38.3 | 20.0 | 569 | 39.60 | 13.70 | 33.62 | 13.08 | 803 |
| KSA31 | 33.23 | 16.16 | 35.69 | 14.93 | 978 | 32.53 | 20.24 | 32.53 | 14.70 | 415 | 25.1 | 16.0 | 39.2 | 19.7 | 589 | 33.37 | 11.21 | 40.97 | 14.45 | 803 |
| KSA32 | 32.72 | 16.62 | 35.49 | 15.18 | 975 | 33.41 | 20.58 | 32.20 | 13.80 | 413 | 27.1 | 15.9 | 37.7 | 19.3 | 554 | 39.88 | 13.88 | 33.63 | 12.63 | 800 |
| KSA33 | 32.93 | 16.21 | 36.05 | 14.80 | 993 | 33.57 | 20.05 | 31.88 | 14.49 | 414 | 26.5 | 16.2 | 37.5 | 19.8 | 550 | 38.83 | 14.02 | 34.12 | 13.03 | 806 |

| Genotypes | atpb-rbcl | | | | | TrnL-TrnF | | | | | TrnL | | | | | TrnT-TrnL | | | | |
|---|---|---|---|---|---|---|---|---|---|---|---|---|---|---|---|---|---|---|---|---|
| | T(U) | C | A | G | Total | T(U) | C | A | G | Total | T(U) | C | A | G | Total | T(U) | C | A | G | Total |
| KSA34 | 32.28 | 16.65 | 36.06 | 15.02 | 979 | 33.50 | 19.90 | 31.80 | 14.81 | 412 | 26.0 | 16.2 | 38.4 | 19.4 | 573 | 38.64 | 13.73 | 34.26 | 13.37 | 823 |
| KSA35 | 32.67 | 16.04 | 36.16 | 15.14 | 1004 | 32.60 | 19.95 | 32.85 | 14.60 | 411 | 26.7 | 15.7 | 38.1 | 19.5 | 554 | 39.29 | 13.92 | 33.62 | 13.18 | 812 |
| KSA36 | 33.54 | 15.90 | 35.28 | 15.28 | 975 | 31.66 | 19.36 | 33.94 | 15.03 | 439 | 26.5 | 16.0 | 37.9 | 19.6 | 551 | 39.41 | 13.63 | 34.32 | 12.64 | 807 |
| KSA37 | 32.99 | 15.73 | 35.96 | 15.32 | 979 | 32.50 | 19.09 | 32.73 | 15.68 | 440 | 27.1 | 15.5 | 38.7 | 18.8 | 595 | 39.70 | 13.52 | 34.37 | 12.41 | 806 |
| KSA38 | 32.96 | 15.61 | 36.14 | 15.30 | 974 | 33.49 | 20.48 | 31.33 | 14.70 | 415 | 27.1 | 15.9 | 37.7 | 19.3 | 584 | 40.08 | 14.03 | 34.13 | 11.76 | 791 |
| KSA39 | 33.03 | 15.64 | 36.40 | 14.93 | 978 | 32.94 | 20.56 | 31.31 | 15.19 | 428 | 27.3 | 15.7 | 37.8 | 19.3 | 535 | 39.41 | 13.88 | 33.46 | 13.26 | 807 |
| KSA40 | 32.39 | 16.47 | 35.21 | 15.93 | 923 | 33.01 | 19.86 | 32.54 | 14.59 | 418 | 26.4 | 15.7 | 38.1 | 19.9 | 554 | 39.18 | 13.84 | 33.37 | 13.60 | 809 |
| KSA41 | | | | | | 33.49 | 19.95 | 32.80 | 13.76 | 436 | 26.6 | 14.9 | 38.2 | 20.3 | 523 | 39.67 | 14.11 | 33.75 | 12.47 | 794 |
| KSA42 | | | | | | 34.49 | 20.14 | 30.79 | 14.58 | 432 | 26.4 | 15.5 | 37.3 | 20.8 | 576 | 40.05 | 14.30 | 33.58 | 12.06 | 804 |
| KSA43 | | | | | | 33.82 | 20.19 | 31.63 | 14.36 | 411 | 27.0 | 14.7 | 37.9 | 20.4 | 530 | 39.88 | 13.75 | 33.75 | 12.63 | 800 |
| KSA44 | 31.89 | 16.76 | 35.57 | 15.78 | 925 | 31.65 | 19.50 | 31.19 | 17.66 | 436 | 26.8 | 15.9 | 37.7 | 19.6 | 560 | 39.30 | 13.81 | 33.21 | 13.68 | 804 |
| KSA45R | 32.86 | 15.84 | 35.68 | 15.62 | 922 | 32.06 | 19.38 | 33.01 | 15.55 | 418 | 27.2 | 15.7 | 37.2 | 20.0 | 541 | 39.60 | 13.82 | 33.50 | 13.08 | 803 |
| KSA46 | 32.86 | 16.43 | 35.61 | 15.10 | 980 | 32.64 | 19.44 | 32.18 | 15.74 | 432 | 26.7 | 15.8 | 37.3 | 20.2 | 544 | 39.35 | 14.16 | 33.58 | 12.91 | 798 |
| KSA47 | 33.10 | 15.68 | 35.95 | 15.27 | 982 | 33.17 | 19.61 | 32.93 | 14.29 | 413 | 26.1 | 16.0 | 38.2 | 19.7 | 563 | 39.63 | 14.13 | 33.50 | 12.75 | 800 |
| KSA48 | 32.39 | 16.30 | 35.65 | 15.65 | 920 | 33.63 | 19.41 | 32.05 | 14.90 | 443 | 28.6 | 21.6 | 32.4 | 17.4 | 574 | 39.63 | 14.16 | 33.66 | 12.55 | 805 |
| KSA49 | 32.65 | 16.05 | 35.79 | 15.51 | 922 | 32.61 | 20.86 | 31.89 | 14.63 | 417 | 26.9 | 17.7 | 35.5 | 19.9 | 583 | 39.30 | 13.93 | 33.58 | 13.18 | 804 |
| KSA50 | 32.68 | 16.02 | 35.82 | 15.48 | 924 | 32.27 | 21.59 | 30.00 | 16.14 | 440 | 26.9 | 15.4 | 37.9 | 19.8 | 551 | 39.78 | 13.97 | 33.29 | 12.97 | 802 |
| KSA51R | 32.39 | 15.71 | 36.51 | 15.38 | 923 | 33.58 | 19.95 | 32.12 | 14.36 | 411 | 26.5 | 16.5 | 37.9 | 19.1 | 570 | 39.46 | 13.93 | 33.91 | 12.70 | 811 |
| KSA52 | 32.68 | 16.07 | 35.40 | 15.85 | 921 | 32.85 | 20.68 | 31.39 | 15.09 | 411 | 26.7 | 15.7 | 37.1 | 20.5 | 536 | 38.75 | 14.02 | 33.58 | 13.65 | 813 |
| KSA59 | 32.36 | 16.13 | 35.71 | 15.80 | 924 | 33.50 | 19.75 | 31.50 | 15.25 | 400 | 26.3 | 15.6 | 38.5 | 19.6 | 556 | 38.27 | 13.49 | 33.54 | 14.70 | 823 |
| KSA60 | 32.50 | 16.36 | 35.75 | 15.38 | 923 | 33.64 | 20.23 | 30.45 | 15.68 | 440 | 26.7 | 16.2 | 37.6 | 19.6 | 551 | 38.56 | 14.20 | 33.78 | 13.46 | 817 |
| KSA61 | 32.29 | 15.87 | 35.64 | 16.20 | 926 | 32.70 | 20.14 | 32.46 | 14.69 | 422 | 25.9 | 16.6 | 37.0 | 20.5 | 595 | 39.23 | 14.20 | 33.50 | 13.08 | 803 |
| KSA62 | 32.86 | 16.02 | 36.22 | 14.90 | 980 | 32.00 | 18.00 | 35.50 | 14.50 | 400 | 27.4 | 15.6 | 36.9 | 20.0 | 544 | 39.70 | 13.86 | 33.58 | 12.86 | 801 |
| KSA63 | 33.09 | 15.93 | 35.96 | 15.02 | 979 | 32.81 | 19.82 | 32.21 | 15.16 | 426 | 26.4 | 15.9 | 37.7 | 19.9 | 552 | 39.57 | 13.43 | 34.05 | 12.95 | 834 |
| Avg. | 33.15 | 16.60 | 34.49 | 15.76 | 963.8 | 32.83 | 19.81 | 32.21 | 15.13 | 425.82 | 26.7 | 15.9 | 37.6 | 19.8 | 566 | 39.18 | 13.88 | 33.84 | 13.10 | 812 |

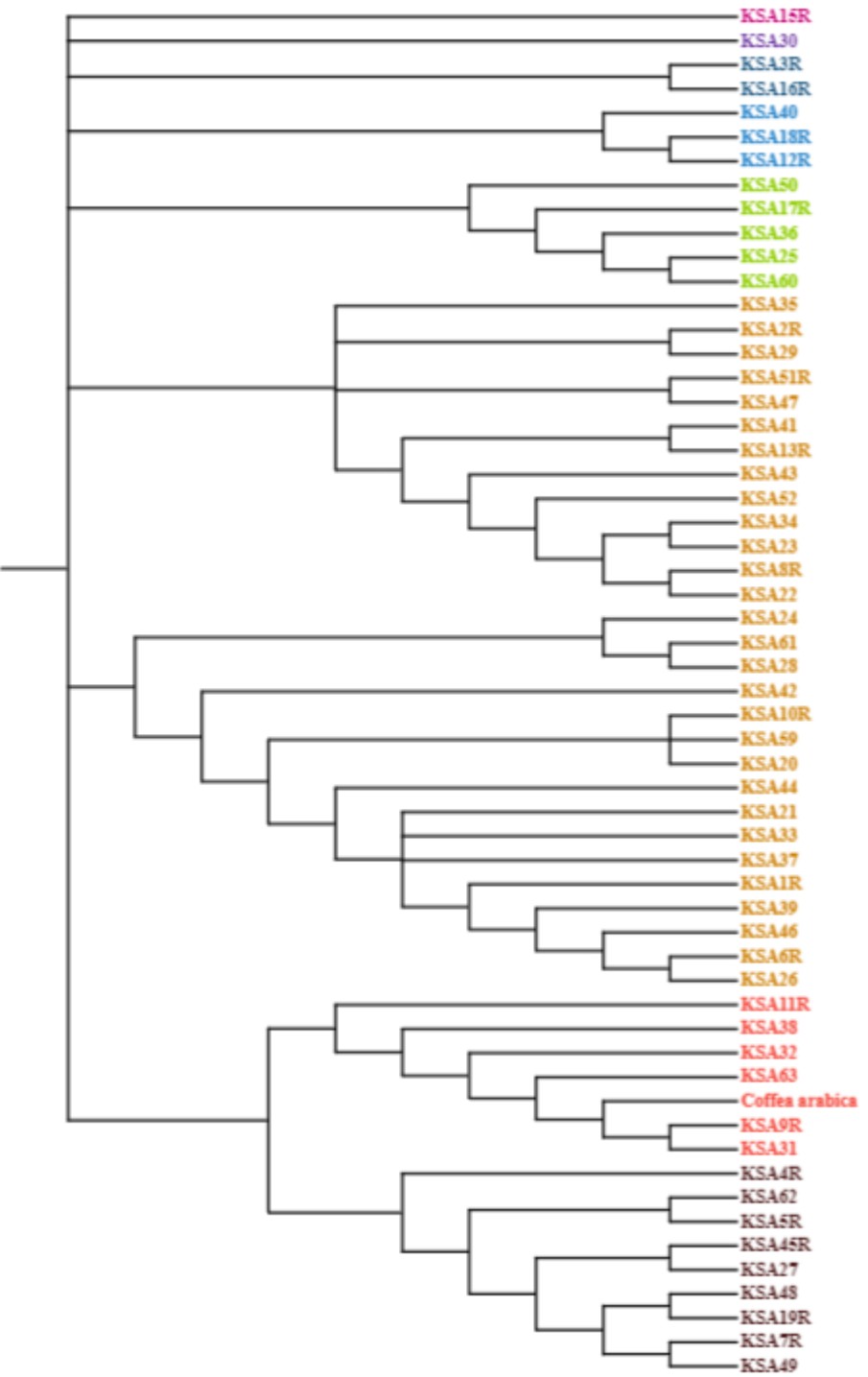

**Figure 1** Evolutionary analysis by maximum likelihood method using four barcodes with 1,000 bootstraps constructed in MEGA 10.0 using the concatenated sequence of atpB-rbcl, trnL, trnT-trnL and trnL-trnF.

Food and beverages adulteration is another widespread malpractice of concern to both traders and consumers. In particular, coffee adulteration aims to mitigate the effects of high prices, product shortages, or reduce production expenses (*Flores-Valdez et al., 2020*). Therefore, there is a real need to develop methods and models for detecting and quantifying coffee adulterants commonly used in coffee.

It is estimated that approximately 60% of wild coffee species are at risk of extinction worldwide. Similarly, underutilized old varieties are disappearing from the orchards. This it underscores the pressing importance of preserving these species through both *in situ* and *ex situ* measures to safeguard their genetic diversity for future use.

While morphological descriptors are commonly used to characterize different coffee species, molecular markers are considered more efficient in distinguishing closely related species and cultivars (*Mishra, Jingade & Huded, 2022*). They are also more precise and reliable than morphological and biochemical markers (*Hao et al., 2009*). Furthermore, several studies have demonstrated that specific regions of the chloroplast genome can serve as DNA barcodes for a wide variety of plant species (*Skuza et al., 2019*; *Meena et al., 2020*). Selection of suitable plastid genomes offers sufficient genetic information for distinguishing between genotypes. Additionally, when choosing suitable DNA barcoding loci, the variable regions should be given a primary consideration (*Mahadani & Ghosh, 2014*). Therefore, the objective of this study was to identify fifty-six local Arabica coffee accessions in the southwestern Saudi Arabia and to evaluate the evolutionary and phylogenetic relationships among them by utilizing four DNA barcoding markers (atpB-rbcl, trnL-trnF, trnL, and trnT-trnL). This research aimed to investigate the potential of four DNA barcode loci(specifically, atpB-rbcL, TrnL, TrnL-trnF, and trnL-trnT from the chloroplast region) for the identification and provision of phylogenetic information on local *Arabica* coffee genotypes. All four regions were successfully amplified using universal primers, yielding clear and reliable results. However, earlier studies have indicated that there were cases of partial amplification from the respective barcode loci's using universal barcode primers (*Hamon et al., 2017*; *Wu et al., 2021*). Similarly, other studies have shown 100% success rate for PCR amplification and sequencing for mangrove (*Guyeux et al., 2019*), duckweeds (*Meena et al., 2020*), and *Coffea* (*Taberlet et al., 1991*). The PCR amplification and sequencing of rbcL fragments in core barcodes of mangrove DNA samples achieved a 100% success rate. Our results demonstrated higher universality and success rates compared to *Kress et al. (2009)* and were consistent with *Pei et al. (2015)*, where success rates ranged from 90% to 100% in forest plant communities within tropical and subtropical regions.

Similarly, other studies (*Vickers, 2017*; *Wu et al., 2019b*) have indicated that additional barcode primers, including matK, rbcL, and trnL-trnF, have demonstrated successful amplification within coffee species. However, no significant differences were recorded in the rate of coffee identification between rbcL + trnH-psbA and other combinations of random fragments, which aligns with the findings of the present study using all four barcodes for genotype identification.

Despite the abundance of available data on DNA barcoding of angiosperms, there is currently limited information regarding specific barcodes that can guarantee an accurate species identification in all cases (*Weigand et al., 2019*). Often, a barcode that performs

effectively for one group of plants may prove inadequate for another group, especially in the case of recently diverged species (*Li et al., 2015*). The current study successfully identified all fifty-six accessions as *Coffea arabica*, except KSA2, KSA41, KSA42 and KSA43 for atpB-rbcl, showcasing the effectiveness of the universal DNA barcode primers. Likewise, multiple studies have extensively documented the reliability of matK and rbcL, either individually or in combination, as DNA barcodes that can be used with confidence across various plant species (*Carneiro de Melo Moura et al., 2019*). Several reports have recommended the utilization of rbcL as a valuable DNA barcode locus, primarily due to its relatively compact length of 500 bp, high success rate of PCR amplification, and excellent sequencing quality (*Wu et al., 2019a*; *Wu et al., 2019b*; *Hong et al., 2022*). However, other DNA barcodes, such as trnL-trnF and the trnL spacer, have also been suggested as reliable alternative barcodes for identification of species (*Kang, 2021*). The extent of sequence variation among the species or terminals under analysis is a crucial factor in determining the effectiveness of any barcoding locus (*Carneiro de Melo Moura et al., 2019*).

The number of singleton variable sites was found to be higher in trnL, trnL-trnF, and trnL-trnT compared to atpB-rbcl. Similarly, trnL and atpB-rbcl had more parsimony information sites than the rbcL barcode spacer region. These findings are consistent with a previous study by *Mishra, Jingade & Huded (2022)*, which reported that trnL-trnF and matK barcodes exhibited greater variability than rbcl in Indian *C. arabica* genotypes. The present study also found similar results for PIS among the four barcodes analyzed. Similarly, previous research has indicated that trnL-trnF and matK loci exhibit greater sequence polymorphism than rbcL, as suggested by *Kimura (1980)* and *Kumar et al. (2018)*. The current study's results support these findings. Hence, the present study found that all four barcode sequences, which were evaluated as candidate barcode markers, met the DNA barcoding criteria outlined by *Li et al. (2015)*. Specifically, these markers exhibited sufficient sequence variability to enable effective discrimination among the Saudi coffee genotypes.

The phylogenetic analysis grouped the Saudi *C. arabica* genotypes into four groups with a clear influence of geographic origin suggesting the genotypes of each region share one or more common ancestor (Fig. 1). For instance, accessions KSA11R, KSA41, KSA42 and KSA43 from the isolated Rayda district of Assir region were grouped in clusters I and II. The accessions representing very old trees (KSA36, KSA44, KSA46, KSA47) segregated in the middle of the phylogenic tree in groups III and IVa. Similar results were reported by *Mishra, Jingade & Huded (2022)* where the grouping using single and multi-locus barcode primers was strongly influenced by the geographic origin of the genotypes. A molecular analysis of coffee genotypes from Saudi Arabia using SRAP markers grouped them into five distinct groups based mostly on their geographic origin (*Al-Ghamedi et al., 2023*). The accessions collected from Jazan region primarily clustered in groups II and IV, whereas those from Al-Baha and Assir regions formed a different group. Similar surveys of genetic diversity among coffee populations in northern Yemen (*Montagnon et al., 2021*) and southern Yemen (*Eskes, 1989*) found that each district (valley) have its own cultivars. Another study using genotyping by sequencing (GBS) showed that genetic closeness correlated with geographic proximity (*Hamon et al., 2017*). The current study provides further evidence

to support this finding. It was also suggested that chloroplast sequences provide more insights into species evolution because they are more conserved (*Guyeux et al., 2019*). For future studies on this economically significant crop, we recommend using sequencing and genome-wide association studies (GWAS) to discover additional polymorphic markers associated with important agro-morphological traits. These markers would be beneficial for a range of investigations in *Coffea*. Ultimately, the polymorphic markers established and confirmed in this research hold potential as a valuable genomic asset for molecular breeding, genotype identification, and biogeography studies on Arabica coffee.

## CONCLUSION

To summarize, this study utilized a DNA barcoding approach to investigate and identify the molecular relationships among fifty-six Arabica coffee accessions collected from the southern region of Saudi Arabia. The three-barcode regions, namely trnT-trnL, trnL-trnF, and trnL, exhibited a higher sequence variability compared to the atpB-rbcl barcode region and effectively differentiated the local coffee genotypes by the presence of unique variable sites (singletons and parsimony). Moreover, the combination of DNA sequences from these barcode loci analyzed using the maximum likelihood phylogenetic method grouped similar coffee genotypes together, providing improved resolution and a better understanding of the population structure. These findings will contribute to future research on the characterization and conservation of Arabica coffee germplasm using DNA barcoding markers.

### Funding

The Deputyship for Research & Innovation, Ministry of Education in Saudi Arabia has funded this research work through project number 1.092. The funders had no role in study design, data collection and analysis, decision to publish, or preparation of the manuscript.

### Grant Disclosures

The following grant information was disclosed by the authors:
The Deputyship for Research & Innovation, Ministry of Education in Saudi Arabia.

### Competing Interests

The authors declare there are no competing interests.

### Author Contributions

- Habib Khemira conceived and designed the experiments, performed the experiments, analyzed the data, prepared figures and/or tables, authored or reviewed drafts of the article, funding search, and approved the final draft.
- Mosbah Mahdhi conceived and designed the experiments, performed the experiments, analyzed the data, prepared figures and/or tables, authored or reviewed drafts of the article, funding search, and approved the final draft.

- Muhammad Afzal analyzed the data, prepared figures and/or tables, authored or reviewed drafts of the article, and approved the final draft.
- Mohammed D.Y. Oteef conceived and designed the experiments, performed the experiments, prepared figures and/or tables, authored or reviewed drafts of the article, funding search, and approved the final draft.
- Taieb Tounekti conceived and designed the experiments, performed the experiments, analyzed the data, prepared figures and/or tables, authored or reviewed drafts of the article, funding search, and approved the final draft.
- Zarraq AL-Faifi conceived and designed the experiments, performed the experiments, analyzed the data, authored or reviewed drafts of the article, funding search, and approved the final draft.
- Wail Alsolami conceived and designed the experiments, performed the experiments, analyzed the data, prepared figures and/or tables, authored or reviewed drafts of the article, funding search, and approved the final draft.

## DNA Deposition

The following information was supplied regarding the deposition of DNA sequences:

Sequences are available at NCBI:

OQ953999–OQ954054

## Data Availability

The data are available in the Supplemental File.

## Supplemental Information

Supplemental information for this article can be found online at http://dx.doi.org/10.7717/peerj.16486#supplemental-information.

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
