# Peer review of "Assessment of genetic diversity and phylogenetic relationship of local coffee populations in southwestern Saudi Arabia using DNA barcoding"

_PeerJ, doi:10.7717/peerj.16486_

## Round 0.1 · original submission · Minor Revisions

Please revise the article considering the comments of reviewers.

Reviewer 1 ·

Basic reporting

• English used in the review is clear and professional in all sections except 'results". Please review it
again and make it in clear English language.
• Figures are relevant and clear.
• Line number 57, please cite the reference.

Experimental design

• Research question is well defined.
• Corrections in Line 255 and 256. Give separate comparison for NSS and NNS.
• Make corrections from line 255-261 as there are mistakes in writing when compared to Table no. 4.
• In line 281, 6 accessions are written but in bracket 7 names are mentioned. Please do check and make
it correct.

Validity of the findings

• All the data provided are robust and statistically sound.
• Conclusions are well linked with the research question.

Additional comments

"Results" section can be explained in some more better way. The comparison portion of results should be once checked carefully.

·

Basic reporting

The English used in the article is good, but a few sentences are unclear mentioned below, and should be addressed.
Line 101-105; Rewrite the sentence for better understanding.
Line 119-124; Split the sentence and rewrite the second sentence for better understanding.
In introduction lines 55-56; the production data is available for the more recent years that can be incorporated and the area of production can also be added.
The figures and data supplied are satisfactory and useful for future experiments in the coffee improvement programs.

Experimental design

This is the original research and well suited to the current situation.
The question raised related to adulteration in coffee is very much relevant, and the discussion should be added.
The methodology described is informative and clear, collection of plant material is appreciable; some corrections should be addressed as mentioned below-
Line 170; What do you mean by “Cooler”?
Line 219; PIC is not an appropriate abbreviation for parsimony informative sites.
Line no 234; “There was a total of 5381 positions in the final dataset”- specify the positions.
Line 249; This should be part of the materials and methods.

Validity of the findings

Sufficient data have been provided to justify the research question and well discussed; the conclusions are satisfying the research problem. The below-mentioned corrections should be incorporated.
Line 255-261; Please correlate the data with Table 4 and describe accordingly.
Line 263; Data represented from Table 5 not from Table 4, make corrections accordingly.
Lines 263-266; mention the average nucleotide composition of trnL-trnF barcode primer.
Line 270; Write PIS grand total value for trnL.
Line 362; Make correction, and replace PIC with PIS.
Lines 85-89 of the introductory paragraph should also be discussed under the heading of “discussion”.
Table 5; Make a correction in the head row, and replace trnC with trnL.
Table S1; Make corrections, and replace PIC with PIS.

Additional comments

Overall work is impactful, check for minor grammar mistakes at your end.

---

## Round 0.2 · Minor Revisions

Please revise the article considering the reviewers comments.

Reviewer 1 ·

Basic reporting

• Corrections are done as suggested.

Experimental design

• In line no. 247, the abbreviation NSS should be replaced with NNS as this comparison is for NNS (No. of nucleotide sites). (Go through Table 4).
• Line no. 248-252 should be removed as it is repeating.
• Add comparison for NH (No.of haplotypes).
• Rest, corrections has been made properly.

Validity of the findings

• Problems are properly justified.

·

Basic reporting

English used meets the standard, the literature reviewed is satisfactory and the necessary corrections have been incorporated.

Experimental design

The article represents originality, the methodology adopted is appropriate and the recommended actions have been well satisfied.

Validity of the findings

Minor corrections suggested previously have been justified.

---

## Round 0.3 · accepted · Accept

Revised article accepted.